**Data Availability Statement:** Data cannot be shared publicly because of restrictions due to Norwegian regulations. Since informed consent specifically on publication of individual data was

# No impact of gestational diabetes mellitus on pregnancy complications in women with PCOS, regardless of GDM criteria used

Stine Lyngvi Fougner[1,2]*, Eszter Vanky[2,3], Tone Shetelig Løvvik[2,3©], Sven Magnus Carlsen[1,2©]

1 Department of Endocrinology, Clinic of Medicine, St. Olavs Hospital, Trondheim University Hospital, Trondheim, Norway, 2 Department of Clinical and Molecular Medicine, Norwegian University of Science and Technology (NTNU), Trondheim, Norway, 3 Department of Obstetrics and Gynaecology, St. Olavs Hospital, Trondheim University Hospital, Trondheim, Norway

© These authors contributed equally to this work.
* stine.fougner@ntnu.no

## Abstract

Polycystic ovary syndrome (PCOS) is characterized by the presence of insulin resistance, and women with PCOS have high prevalence of gestational diabetes (GDM). Both conditions have been associated with increased risk for pregnancy complications such as preterm birth, preeclampsia and increased offspring birth weight. We aimed to estimate the prevalence of GDM in women with PCOS using both previous and new diagnostic criteria, and to analyse whether the risk of pregnancy complications increased with the presence of GDM. In addition, we aimed to assess the response to metformin treatment in PCOS women with GDM. We performed post-hoc analysis of three prospective, double blinded studies of altogether 791 pregnant women with PCOS randomized to either metformin or placebo treatment from first trimester to delivery. Glucose data allowing GDM classification after previous (WHO 1999) and new (WHO 2013 and Norwegian 2017) diagnostic criteria were available for 722 of the women. Complications such as preeclampsia, late miscarriage and preterm birth, birth weight and gestational age were correlated to the presence of GDM and metformin treatment. The prevalence of GDM was 28.3% (WHO 1999), 41.2% (WHO 2013) and 27.2% (Norwegian 2017). Having GDM already in first trimester associated with increased risk for late miscarriage (p<0.01). Having GDM according to newer criteria correlated to increased maternal age and BMI (p<0.001). Otherwise, having GDM (any criteria) correlated neither to the development of preeclampsia, nor to birth weight z-score or the proportion of offspring being large for gestational weight. Maternal age and BMI, parity and gestational weight gain, but not GDM or metformin treatment, were determinants for birth weight z-score. Conclusion: in pregnant women with PCOS, having GDM did not increase the risk for other pregnancy complications except for an increased risk for late miscarriage among those with GDM already in the first trimester.

not requested from the participants before inclusion, it is not allowed by the Norwegian regulations to publish individual data. This has been discussed with the leader of the Regional Committee for Health Research Ethics prior to submission and is in accordance with Norwegian national legislation, paragraph 20. Contact information, Regional Committee for Health Research Ethics of Central Norway: rek-midt@mh.ntnu.no.

**Funding:** The pilot study had no external funding except for salary costs to the primary investigator (EV) from her institution (St. Olav's University Hospital and the Norwegian University of Science and Technology). For the pilot study and the PregMet study, Metformin and placebo tablets were delivered free of charge by Weifa A/S, Oslo, Norway. The PregMet study also received grants from the Liaison Committee between the Central Norway Regional Health Authority and the Norwegian University of Science and Technology. The PregMet2 study received funding from The Research Council of Norway. None of the funders had any role in the study design, data collection and analysis, decision to publish, or the preparation of the manuscript.

**Competing interests:** The authors have declared that no competing interests exist.

## Introduction

Gestational diabetes mellitus (GDM) is defined as glucose intolerance diagnosed in pregnancy. In most women with GDM, β-cell dysfunction due to chronic insulin resistance is present prior to pregnancy. The physiological insulin resistance seen in the second half of normal pregnancies adds to this chronic insulin resistance, leading to elevated glucose levels. Hence, factors increasing insulin resistance, such as advanced maternal age and increased body mass index (BMI), are known risk factors for the development of GDM [1,2]. According to the International Diabetes Federation (IDF), one out of seven pregnancies (14%) worldwide meets the WHO 2013 criteria for GDM (IDF Diabetes Atlas 2017).

Polycystic ovary syndrome (PCOS) is a common endocrine disorder in women of fertile age, with a prevalence of up to 20% in European studies using the Rotterdam 2003 diagnostic criteria for PCOS [3]. In the non-pregnant state, most women with PCOS are characterized by the presence of both hyperandrogenemia and increased insulin resistance [4]. Insulin resistance has been found to be present in 44–70% of women with PCOS, depending on the method used [5]. A meta-analysis of clamp studies demonstrated an overall 27% reduction of insulin sensitivity in PCOS women compared to controls. Importantly, the insulin sensitivity was reduced also in lean women with PCOS compared to lean controls [6]. The risk for developing GDM is therefore expected to be increased in women with PCOS. A recent meta-analysis of 40 studies including 17 800 PCOS pregnancies found a relative risk of 2.78 for GDM compared to non-PCOS pregnancies. This increased risk was particularly among non-obese women [7]. However, another review demonstrated substantial heterogeneity among studies, suggesting that the results could depend on differences in BMI and study designs [8]. Two Nordic prospective cohort studies and review did not find any increased risk for GDM in PCOS *per se*, but rather that the risk of GDM depended on factors like ethnicity, BMI, age and the diagnostic criteria for GDM used [9,10].

Women with PCOS are also at an increased risk of developing other pregnancy complications such as preterm birth, hypertension and preeclampsia [7]. A recent study concluded, however, that the rate of perinatal complications such as preeclampsia and GDM was not increased in pregnant women with PCOS after adjusting for gestational weight gain [11]. Another study of pregnant women with GDM found that having PCOS in addition further increased the risk for pregnancy complications [12].

In the general pregnant population, GDM has been associated with increased risk for both maternal and foetal complications such as preeclampsia, preterm delivery, macrosomia and large for gestational age (LGA) babies [13,14]. The HAPO study demonstrated a positive association between glucose levels and the proportion of neonates with a birth weight above 4000 g, with no obvious cut-off level, but rather a continuous rise throughout the total glucose level range of both fasting and 1 and 2 hours after an oral glucose load [15]. Gestational weight gain was not registered in this observational study.

Given the increased prevalence of both GDM and other pregnancy complications in women with PCOS, we hypothesized that the increased risk for complications in pregnancies with PCOS is related to common underlying risk factors in this population and not to the concomitant presence of GDM *per se*. In this study, we present the prevalence of GDM in a large, pooled patient cohort from three randomized controlled studies of pregnant women with PCOS [16–18], using both former and present criteria for GDM. Further, we compared the risk of maternal and neonatal complications in PCOS women with and without GDM and examined whether treatment with metformin had any impact on pregnancy complications in women with GDM.

## Patients and methods

### Patients, randomization and treatment

The patients included in this post hoc analysis are pregnant women with PCOS who took part in three prospective, randomized, double blinded, placebo-controlled trials with similar designs. The first study was a pilot study of 40 women, all at St. Olavs Hospital, Trondheim University Hospital, Norway. The PregMet and PregMet2 studies were multicentre trials. PregMet included 273 women at eleven secondary study centres in Norway, while PregMet2 included 478 women in fourteen secondary study centres in Norway, Sweden and Iceland. The participants were randomized to treatment with either metformin or placebo. The target dose of metformin was 1700 mg daily in the pilot study and 2000 mg daily in the PregMet and Preg-Met2 studies. Women diagnosed with GDM were referred according to local guidelines for further assessment and treatment, without interfering with the study medication. The three studies are described in detail elsewhere [16–18].

According to the protocols, a 75 g oral glucose tolerance test (OGTT) was performed at inclusion in the first trimester ($\leq$ week $12^{+6}$), and then in week 19 and 32 (Pilot and PregMet) or in week 28 (PregMet2) in the patients not diagnosed with GDM after the initial OGTT. However, for some patients the OGTT was omitted at one or several visits, due to reasons such as nausea, previous bariatric surgery, failing willingness, analysis failure or non-fasting participants. In this post hoc analysis of GDM in women with PCOS, only women with sufficient available data on glucose metabolism were included. Of the 791 women completing one of these trials, an OGTT had been performed at inclusion in 772 (97.6%) of the women. For evaluation of the impact of GDM in early pregnancy, these 772 women were included. However, for the rest of this post hoc study we included only women who had an OGTT either at week 28 or 32, or at the last visit prior to delivery, unless a fasting glucose alone or the OGTT at an earlier visit led to the diagnosis of GDM. In total, 722 women (91.3%) were included in the main analyses.

### GDM classification

In the study protocols for all three studies, GDM was diagnosed in accordance with the WHO 1999 criteria. The new WHO 2013 criteria for diagnosing GDM were released during the Preg-Met2 study period, but this was implemented only in Iceland at the end of the study period. Hence, the great majority of patients were diagnosed and treated in accordance with the WHO 1999 criteria. In the present study, we classified GDM using three different criteria: (i) the WHO 1999 criteria (fasting plasma glucose (FPG) $\geq$7.0 mmol/L or 2 h plasma glucose $\geq$7.8 mmol/L), (ii) the WHO 2013 criteria (FPG 5.1–6.9 mmol/L or 2 h plasma glucose 8.5–11.0 mmol/L) based on the 1.75 SD in the HAPO study and (iii) the Norwegian 2017 criteria (FPG $\geq$ 5.3 mmol/L or 2 h plasma glucose $\geq$9.0 mmol/L) based on the 2.0 SD in the HAPO study [15,19].

### Baseline data

Baseline data were recorded at the inclusion visit in the first trimester of pregnancy, and included maternal weight and age, parity, smoking status and comorbidities. BMI was calculated from maternal weight and height as kg/m$^2$. A 75 g oral glucose tolerance test (OGTT) was also performed at inclusion.

### Pregnancy outcome

All late miscarriages (week $13^{+0}$ to $22^{+6}$) and preterm births (week $23^{+0}$ to $36^{+6}$) were recorded, as was the development of hypertension and preeclampsia during pregnancy. Maternal weight

gain from inclusion until the gestational week 36, was recorded for the women attending the visit in week 36.

When glycaemic targets were not achieved through diet and lifestyle modifications, insulin treatment was initiated. The decision to initiate insulin treatment was based on local guidelines and practice, and not stated in the study protocols. For PregMet2, the practice for clinical GDM classification and criteria for insulin treatment changed in Iceland during the study period, but not in Norway or Sweden.

### Neonatal outcome

Birth weight, gender and gestational age were recorded. For all live births after gestational week 24, a gestational age- and gender-adjusted birth weight z-score was calculated based on Niklasson's standard values from a large Swedish population [20]. The z-scores express the deviation between observed values and the Swedish population mean birth weight, adjusted for sex and gestational age at birth. Small for gestational age (SGA) was defined as birth weight <10 percentile of the gestational age (z-score <-1.28), and large for gestational age (LGA) as a birth weight >90 percentile of the gestational age (z-score >1.28).

### Statistical analyses

All analyses were performed using SPSS version 25 (IBM SPSS, Armonk, NY, USA). Comparisons between groups were performed with t-test for independent samples. Chi square test was used to analyse differences in proportions of dichotomized variables. Pearson correlation test was used to analyse correlation between continuous variables. Linear regression analysis was used to analyse the determinants of birth weight, and logistic regression analysis for analysis of determinants for late miscarriage. In the multiple regression, we included the parameters that either were significantly correlated with the outcome in the univariate analysis or were considered clinically relevant. For related variables, such as maternal weight and BMI or glucose data and GDM status, only one was included.

To adjust for multiple analyses, p-values ≤0.01 were considered significant, while p-values between 0.01 and 0.05 were regarded as a trend.

### Ethics

The studies were approved by the Regional Committees for Medical Research Ethics (all three studies; project number 51–2000, 145.04 and 2011/1434), the Regional Ethical Review Board in Stockholm, Sweden (PregMet2, Dnmb: 2012/1200-31/2), and the National Bioethics Committee of Iceland (Pregmet2, VSNb2012100011/03.10). All studies were approved by the Medicines Agency in Norway, and for PregMet2 also approved in Sweden and Iceland.

They were conducted according to the Declaration of Helsinki II, and written informed consent was obtained from all patients.

The studies are registered at ClinicalTrials.gov (NCT 03259919, NCT 00159536, NCT 01587378). The first study was registered after the enrolment of participants because the study started in 2000, prior to the regular use of this web registry. The authors confirm that all ongoing and related trials for this intervention are registered.

## Results

### GDM and baseline characteristics

Of the 722 women with sufficient glucose data, 706 women could be classified in accordance with the WHO 1999 GDM criteria, 702 women by using the WHO 2013 criteria and 685

women by using the Norwegian 2017 criteria. Of the 706 women classified using the WHO 1999 criteria, 200 women (28.3%) had GDM. Using the WHO 2013 criteria, 289 out of 702 women (41.2%) had GDM, while 186 out of 685 women (27.2%) had GDM according to the Norwegian 2017 criteria.

Women with GDM according to WHO 1999 criteria tended to be older than women without GDM (p = 0.038), while with the newer GDM criteria (WHO 2013 and Norwegian 2017 criteria) a correlation between increasing age and the presence of GDM was found (p<0.01). BMI was significantly higher in women with GDM using all three criteria (p< 0.01). No difference in pre-existing comorbidity was found between women with and without GDM (47% vs 50%, ns) when using the WHO 1999 criteria, while a trend towards increased comorbidity in women with GDM was found using the WHO 2013 criteria (44% vs 53%, p = 0.024) and the Norwegian 2017 criteria (45% vs 54%, p = 0.038). The proportion of women with and without GDM, according to all criteria, was similar among women randomized to metformin and placebo treatment.

Detailed baseline data according to GDM status are presented in Table 1a and 1b (WHO criteria 1999 and 2013) and S1 Table (Norwegian 2017 criteria).

At inclusion in the first trimester, fasting glucose levels correlated to maternal BMI (R = 0.24, p<0.001) and maternal age (R = 0.15, p<0.001), while 2-hour glucose level during the OGTT correlated to maternal BMI (R = 0.17, p<0.001) and tended to correlate with maternal age (R = 0.08, p = 0.031).

## Maternal outcomes

**GDM during any stage of pregnancy.** Women with and without GDM had similar prevalence of preeclampsia and pregnancy-induced hypertension, independent of the GDM criteria used. For all three GDM criteria, women with GDM gained less weight during pregnancy than women without GDM. For details, see Table 1a and 1b (WHO criteria 1999 and 2013) and S1 Table (Norwegian 2017 criteria). During the study, women diagnosed with GDM in accordance with the WHO 1999 criteria were informed of having GDM and advised about lifestyle and diet modifications to treat GDM. When those women were excluded from the analysis, there was no difference in gestational weight gain between women classified with GDM and those without, when the new WHO 2013 criteria were used (11.0 vs 10.2 kg, p = 0.12, n = 477).

The pregnancies with GDM were on average 4 days shorter than the pregnancies without GDM when using the WHO 2013 criteria; however, using the WHO 1999 criteria there was only a tendency for shorter gestational age (Table 1). Again, when excluding the women who received a GDM diagnosis during pregnancy (using the WHO 1999 criteria), there was no difference in gestational age between the women classified with or without GDM when the WHO 2013 criteria were used (276 vs 278 days, p = 0.27).

**GDM at inclusion.** Women diagnosed with GDM already at inclusion in the first trimester had a higher risk of late miscarriage than the women with a normal OGTT, but there was no significant difference in the prevalence of preterm birth (Table 2a). In a logistic regression analysis, only 2-hour glucose level during the first trimester OGTT emerged as a risk factor for late miscarriage, while fasting glucose, smoking habits, metformin treatment, maternal age and BMI did not (Table 2b). Women with late miscarriage had significantly higher 2-hour glucose levels than women without late miscarriage (7.3 ± 1.6 versus 5.8 ± 1.5 mmol/l, p<0.001).

## Neonatal outcomes

When the WHO 1999 criteria were used, offspring born to mothers with GDM had significantly lower birth weight than did offspring born to mothers without GDM (p = 0.008); this

**Table 1. Clinical characteristics and pregnancy outcome for women with and without gestational diabetes (GDM).** a) GDM in accordance with WHO 1999-criteria in 706 women with PCOS. b) GDM diagnosed in accordance with WHO 2013-criteria in 702 women with PCOS.

a) GDM in accordance with WHO 1999-criteria in 706 women with PCOS.

| | | Non-GDM | GDM | P-value |
|---|---|---|---|---|
| | N (%) | 506 (72) | 200 (28) | |
| Randomization | Metformin | 248 (49) | 97 (49) | 0.90 |
| Baseline data | Age, years | 29.6 ± 4.3 | 30.4 ± 4.6 | 0.038 |
| | BMI, kg/m2 | 28.1 ± 6.2 | 29.6 ± 6.5 | **0.006** |
| | Weight, kg | 79.4 ± 18.2 | 81.8 ± 18.0 | 0.12 |
| | Nulliparous | 297 (59) | 107 (54) | 0.23 |
| | Comorbidity | 237 (47) | 99 (50) | 0.52 |
| | Smoking | 26 (5) | 10 (5) | 0.95 |
| Maternal outcome | HT, debut in pregn. | 28 (6) | 6 (3) | 0.09 |
| | Preeclampsia | 34 (7) | 9 (5) | 0.27 |
| | Weight gain, Kg [*] | 10.8 ± 4.9 | 8.2 ± 5.4 | **<0.001** |
| Neonatal outcome | Birth weight, g | 3576 ± 562 | 3445 ± 655 | **0.008** |
| | Birth weight, z-score | -0.01 ± 1.02 | -0.14 ± 1.08 | 0.16 |
| | Gest. age, days | 278 ± 18 | 273 ± 23 | 0.013 |
| | SGA/LGA [$] | 52 (10)/45 (9) | 25 (13)/19 (10) | 0.60 |

b) GDM diagnosed in accordance with WHO 2013-criteria in 702 women with PCOS.

| | | Non-GDM | GDM | p-value |
|---|---|---|---|---|
| | N (%) | **413 (59)** | **289 (41)** | |
| Randomization | Metformin | 197 (48) | 139 (48) | 0.94 |
| Baseline data | Age, years | 29.4 ± 4.2 | 30.3 ± 4.6 | 0.009 |
| | BMI, kg/m2 | 27.2 ± 5.7 | 30.5 ± 6.6 | <0.001 |
| | Weight, kg | 76.6 ± 16.7 | 85.1 ± 19.3 | <0.001 |
| | Nulliparous | 242 (59) | 165 (57) | 0.33 |
| | Comorbidity | 183 (44) | 153 (53) | 0.024 |
| | Smoking | 21 (5) | 15 (5) | 0.96 |
| Maternal outcome | HT, debut in pregn. | 19 (5) | 14 (5) | 0.46 |
| | PE | 24 (6) | 19 (7) | 0.68 |
| | Weight gain, kg [*] | 10.9 ± 4.7 | 9.0 ± 5.6 | <0.001 |
| Neonatal outcome | Birth weight, g | 3539 ± 564 | 3515 ± 680 | 0.60 |
| | Birth weight, z-score | -0.08 ± 1.00 | 0.01 ± 1.07 | 0.28 |
| | Gest. age, days | 278 ± 16 | 274 ± 25 | 0.008 |
| | SGA/LGA | 47 (11)/33 (8) | 29 (10)/29 (5) | 0.56 |

a) Values given as mean ± SD or N (%) as appropriate. [*] N = 647. HT hypertension, PE preeclampsia, SGA small for gestational age, LGA large for gestational age.

b) Values given as mean ± SD or N (%) as appropriate. HT hypertension, PE preeclampsia, SGA small for gestational age, LGA large for gestational age. [*] N = 635.

lower birth weight was not found when the WHO 2013 the Norwegian 2017 GDM criteria were used. However, there was no difference in birth weight z-score or the proportion of SGA and LGA neonates between women with or without GDM using any of the GDM criteria, see Table 1a and 1b (WHO criteria) and S1 Table (Norwegian criteria).

Offspring birth weight z-score showed a positive correlation to maternal weight and BMI at inclusion, parity and maternal weight gain during pregnancy (p<0.001 for all), but only a trend towards correlation with fasting glucose level at inclusion (p = 0.045). In linear regression analysis, maternal age and BMI at inclusion, parity and maternal gestational weight gain during pregnancy correlated to offspring birth weight z-score. GDM according to WHO 1999

**Table 2.** a) The risk of late miscarriage and preterm birth, according to GDM status at inclusion in 772/773 women. b) Determinants of late miscarriage, logistical regression.

| a) The risk of late miscarriage and preterm birth, according to GDM status at inclusion in 772/773 women. | | | | | | |
|---|---|---|---|---|---|---|
| | GDM at inclusion (WHO 1999) | | p-value | GDM at inclusion (WHO 2013) | | p-value |
| | Yes | No | | Yes | No | |
| **N (%)** | 77 (10) | 695 (90) | | 160 (21) | 613 (79) | |
| **Late miscarriage** | 4 (5.2) | 7 (1.0) | **0.003** | 6 (3.8) | 5 (0.8) | **0.005** |
| **Preterm birth** | 3 (3.9) | 44 (6.3) | 0.40 | 7 (4.4) | 40 (6.5) | 0.31 |

b) Determinants of late miscarriage, logistical regression.

| | Late miscarriage | | | | |
|---|---|---|---|---|---|
| | Univariable | | | Multivariable | |
| | OR (95% CI) | p-value | | OR (95% CI) | p-value |
| **Maternal age** | 1.13 (0.99–1.29) | 0.07 | | 1.13 (0.98–1.32) | 0.10 |
| **Maternal BMI** | 1.06 (0.98–1.16) | 0.14 | | 1.06 (0.96–1.16) | 0.24 |
| **Smoking** | 0.0* | 1.00 | | 0.0* | 1.00 |
| **Randomization** | 2.65 (0.70–10.05) | 0.15 | | 2.56 (0.66–10.00) | 0.18 |
| **Fasting glucose, incl**. | 1.31 (0.37–4.6) | 0.67 | | 0.46 (0.12–1.8) | 0.27 |
| **2-hour glucose, incl**. | 1.71 (1.24–2.36) | <0.01 | | 1.69 (1.20–2.39) | 0.003 |
| **Total model (enter)** | | | | | p = 0.009 |

OR = odds ratio, CI = confidence interval.

*Confidence interval cannot be computed, as none of the women with late miscarriage did smoke (Fisher's exact test p = 0.65). Multivariable logistical regression using model: Enter.

Data given as N (%).

or WHO 2013 criteria, randomization to metformin treatment or smoking habits did not correlate to birth weight z-score (Table 3).

## Early versus late GDM

Of the 195 women with GDM classified by the WHO 1999 criteria, 77 women (39%) had GDM already at inclusion in the first trimester; classified by the WHO 2013 criteria, however, as many as 160 of 284 women (56%) were diagnosed with GDM in the first trimester of pregnancy. The women with GDM diagnosed at inclusion were older and received insulin treatment more often during pregnancy than the women who developed GDM later in pregnancy (see Table 3). There were no differences between these GDM groups regarding maternal weight and BMI, parity, comorbidity or smoking. Women with early GDM tended to gain less weight during pregnancy than the women with late GDM (p = 0.015 for WHO 1999 and p = 0.031 for WHO 2013 criteria).

Among women with GDM in accordance with WHO 1999 criteria, early GDM tended to correlate to shorter gestational age (mean 269 vs 276 days, p = 0.032), but this was not found for those classified in accordance with the WHO 2013 criteria. There were no differences in offspring birthweight, birthweight z-score, proportion of SGA and LGA babies (Table 4a and 4b).

## Metformin treatment in women with GDM

When the WHO 2013 criteria were used, women with GDM who were randomized to metformin treatment had fewer late miscarriages and preterm births (combined) than women randomized to placebo treatment (2% versus 12%, p = 0.001), while only a trend was found when

**Table 3.  Determinants of offspring birth weight (z-score) in women with PCOS, linear regression.**

| | Birth weight z-score | | | | | |
|---|---|---|---|---|---|---|
| | Univariable | | Multivariable[1] | | Multivariable[2] | |
| | B | p-value | β (95% CI) | p-value | β (95% CI) | p-value |
| **Maternal age** | -0.03 | 0.40 | -0.13 (-0,5--0,01) | 0.01 | -0.12 (-0.05--0,01) | 0.002 |
| **Maternal BMI** | 0.14 | <0.001 | 0.18 (0.02–0.04) | <0.001 | 0.19 (0.02–0.04) | <0.001 |
| **Maternal weight** | 0.19 | <0.001 | | | | |
| **Smoking** | -0,06 | 0.13 | -0.07 (-0.70–0.14) | 0.06 | -0.07 (-0.69–0.02) | 0.07 |
| **Parity** | 0.22 | <0.001 | 0.27 (0.27–0.50) | <0.001 | 0.28 (0.29–0.51) | <0.001 |
| **Randomization** | 0.00 | 0.93 | -0.00 (-0.16–0.15) | 0.94 | 0.00 (-0.15–0.16) | 0.97 |
| **Fasting glucose, incl** | 0.07 | 0.045 | | | | |
| **2-hour glucose, incl** | 0.03 | 0.46 | | | | |
| **GDM (WHO 1999)** | -0.05 | 0.16 | | | -0.05 (-0.28–0.70) | 0.24 |
| **GDM (WHO 2013)** | 0.04 | 0.28 | 0.05 (-0.05–0.27) | 0.19 | | |
| **Maternal weight gain** | 0.14 | <0.001 | 0.22 (0.03–0.06) | <0.001 | 0.22 (0.03–0.06) | <0.001 |
| **Total** | | | $R^2 = 0.126$ | | $R^2 = 0.131$ | |

[1]GDM after WHO 2013 criterion,

[2]GDM after WHO 1999 criterion.

the other two GDM criteria were used. There were no differences in the prevalence of pre-eclampsia or hypertension. Gestational weight gain was significantly lower in the women treated with metformin than in the placebo group. A tendency was found towards reduced gestational age among the women with GDM treated with metformin, but no difference in off-spring birth weight z-score. Metformin treatment did not affect the proportion of women receiving insulin treatment using any of the GDM criteria. For details, see Table 5 (WHO criteria) and S2 Table (Norwegian criteria).

## Metformin treatment in women without GDM

When the WHO 1999 criteria for GDM were used, metformin treatment was associated with fewer late miscarriages and preterm birth (combined) than placebo treatment in the women without GDM (4% versus 10%, p = 0.008), but not when using the WHO 2013 criteria. A similar trend was found using the Norwegian 2017 GDM criteria. There were no differences in the prevalence of preeclampsia or hypertension. Gestational weight gain was significantly lower in the women treated with metformin than in the placebo group. Gestational age and birth weight z-score were independent of whether the women without GDM were treated with metformin or placebo. For details, see Table 5 (WHO criteria) and S2 Table (Norwegian criteria).

## Discussion

The main findings in this large cohort of pregnant women with PCOS from three randomized placebo-controlled studies on metformin treatment are that: 1) the prevalence of GDM varied markedly with the criteria used, 2) the presence of GDM did not increase the incidence of pregnancy complications, except for an increased risk of late miscarriage in women with GDM already in first trimester, and 3) the use of metformin did not correlate to the presence of GDM, as previously described in this cohort [16]. However, metformin treatment reduced the composite endpoint of late miscarriages and preterm births in PCOS women both with and without GDM. Otherwise, metformin treatment did not correlate to the presence of other maternal or neonatal complications in women with GDM.

**Table 4. GDM diagnosed at inclusion (in first trimester) compared to GDM diagnosed later in pregnancy.** a) GDM after WHO 1999 criteria, N = 195. b) GDM after WHO 2013 criteria, N = 284.

a) GDM after WHO 1999 criteria, N = 195:

| | | GDM | | p-value |
|---|---|---|---|---|
| | | Early GDM | Late GDM | |
| | N (%) | 77 (39) | 118 (61) | |
| **Baseline data** | Age, years | 31.5 ± 4.2 | 29.6 ± 4.6 | 0.004 |
| | Nulliparity | 36 (46) | 66 (55) | 0.10 |
| | BMI, kg/m2 | 30.5 ± 6.2 | 29.0 ± 6.7 | 0.11 |
| | Weight, kg | 83.8 ± 16.8 | 80.5 ± 18.8 | 0.21 |
| | Comorbidity | 32 (42) | 62 (53) | 0.13 |
| | Smoking | 4 (5) | 6 (5) | 0.97 |
| **Maternal outcome** | HT | 1 (1) | 5 (4) | 0.22 |
| | PE | 4 (5) | 5 (4) | 0.76 |
| | Preterm birth | 3 (4) | 6 (5) | 0.65 |
| | Weight gain, kg [*] | 7.0 ± 4.8 | 9.0 ± 5.7 | 0.015 |
| | Gest. age, days | 269 ± 34 | 276 ± 12 | 0.032 |
| | Insulin treatment | 17 (22) | 6 (5) | <0.001 |
| **Neonatal outcome** | Birth weight, g | 3337 ± 793 | 3507 ± 552 | 0.08 |
| | Birth weight, z-score | -0.19 ± 1.09 | -0.10 ± 1.10 | 0.58 |
| | SGA/LGA [$] | 9 (12)/6 (8) | 16 (14)/13 (12) | 0.78 |

b) GDM after WHO 2013 criteria, N = 284:

| | | GDM | | p-value |
|---|---|---|---|---|
| | | GDM at incl | Late GDM | |
| | N (%) | 160 (56) | 124 (44) | |
| **Baseline data** | Age, years | 30.8 ± 4.6 | 29.4 ± 4.4 | **0.008** |
| | Nulliparity | 90 (56) | 72 (58) | 0.61 |
| | BMI, kg/m2 | 31.1 ± 6.4 | 29.9 ± 6.8 | 0.13 |
| | Weight, kg | 86.6 ± 18.4 | 83.4 ± 20.0 | 0.16 |
| | Comorbidity | 83 (52) | 66 (53) | 0.82 |
| | Smoking | 6 (4) | 9 (7) | 0.19 |
| **Maternal outcome** | HT | 7 (4) | 7 (6) | 0.73 |
| | PE | 9 (6) | 10 (8) | 0.41 |
| | Preterm birth | 7 (4) | 9 (7%) | 0.30 |
| | Weight gain, kg [*] | 8.3 ± 5.4 | 9.9 ± 5.7 | 0.031 |
| | Gest. age, days | 271 ± 30 | 276 ± 17 | 0.10 |
| | Insulin treatment | 20 (13) | 4 (3) | **0.005** |
| **Neonatal outcome** | Birth weight, g | 3505 ± 718 | 3518 ± 641 | 0.88 |
| | Birth weight, z-score | 0.03 ± 1.07 | -0.02 ± 1.09 | 0.69 |
| | SGA/LGA [$] | 15 (10)/15 (10) | 14 (11)/14 (11) | 0.80 |

a) Values given as mean ± SD or N (%) as appropriate. HT hypertension, PE preeclampsia, SGA small for gestational age, LGA large for gestational age. [*] N = 175, [$] N = 191.

b) Values given as mean ± SD or N (%) as appropriate. HT hypertension, PE preeclampsia, SGA small for gestational age, LGA large for gestational age. [*] N = 247, [$] N = 277.

## GDM prevalence

In the present study, a large proportion of the women with PCOS were diagnosed with GDM although the prevalence varied markedly with the criteria used. This is in accordance with

**Table 5. Pregnancy complications in patients treated with metformin and placebo, women with and without GDM.**

| | | GDM | | | Non-GDM | | |
|---|---|---|---|---|---|---|---|
| | | **Metformin** | **Placebo** | **p-value** | **Metformin** | **Placebo** | **p-value** |
| **WHO 1999 criteria** | N | 97 | 103 | | 248 | 258 | |
| | **Hypertension** | 3 (3) | 3 (3) | 0.31 | 14 (6) | 14 (6) | 0.92 |
| | **Preeclampsia** | 6 (6) | 3 (3) | 0.26 | 14 (6) | 20 (8) | 0.34 |
| | **Late miscarriage/preterm birth** | 2 (2) | 10 (10) | 0.023 | 10 (4) | 26 (10) | 0.008 |
| | **SGA/LGA** | 9 (9)/9 (9) | 16 (16)/10 (10) | 0.33 | 29 (12)/24 (10) | 23 (9)/21 (8) | 0.47 |
| | **Birth weight, g** | 3527 ± 511 | 3366 ± 761 | 0.08 | 3580 ± 522 | 3571 ± 600 | 0.86 |
| | **Birth weight, z-score** | -0.09 ± 1.05 | -0.18 ± 1.11 | 0.09 | -0.06 ± 1.07 | 0.04 ± 0.96 | 0.27 |
| | **Gest. age, days** | 277 ± 11 | 270 ± 30 | 0.036 | 278 ± 16 | 277 ±20 | 0.45 |
| | **Maternal weight gain, kg** | 6.6 ± 7.6 | 9.2 ± 5.2 | 0.009 | 9.8 ± 4.9 | 11.8 ± 4.7 | <0.001 |
| | **Insulin treatment** | 9 (9) | 14 (14) | 0.339 | 1 (0.4)* | 0 (0) | 0.31 |
| **WHO 2013 criteria** | N | 139 | 150 | | 198 | 215 | |
| | **Hypertension** | 9 (6) | 5 (3) | 0.23 | 8 (4) | 11 (5) | 0.78 |
| | **Preeclampsia** | 9 (6) | 10 (7) | 0.95 | 11 (6) | 13 (9) | 0.83 |
| | **Late miscarriage/preterm birth** | 3 (2) | 18 (12) | 0.001 | 9 (5) | 20 (10) | 0.06 |
| | **SGA/LGA** | 9 (6)/14 (10) | 20 (14)/15 (10) | 0.12 | 26 (13)/17 (9) | 21 (10)/16 (8) | 0.51 |
| | **Birth weight** | 3631 ± 480 | 3406 ± 811 | 0.005 | 3521 ± 533 | 3556 ± 592 | 0.54 |
| | **Birth weight, z-score** | 0.10 ± 1.03 | -0.08 ± 1.11 | 0.17 | -0.17 ± 1.05 | 0.01 ± 0.95 | 0.06 |
| | **Gest. age, days** | 277 ± 19 | 270 ± 30 | 0.022 | 278 ± 11 | 277 ± 20 | 0.49 |
| | **Maternal weight gain, kg** | 8.0 ± 5.4 | 10.0 ± 5.6 | 0.004 | 9.9 ± 4.8 | 11.8 ± 4.4 | <0.001 |
| | **Insulin treatment** | 10 (7) | 14 (9) | 0.51 | 0 | 0 | |

Values given as mean ± SD or N (%) as appropriate. SGA small for gestational age, LGA large for gestational age.

*One patient receiving metformin was treated with insulin due to GDM diagnosed using the new WHO 2013 criteria, but she did not have GDM using the WHO 1999 criteria.

previous reports on GDM in PCOS pregnancies [7]. However, a recent study from Finland concluded that PCOS is not an independent risk factor for GDM, but found that the increased risk mainly was related to adiposity, increased age, heritage for diabetes and maternal preterm birth [9]. Our cohort had no control group of women without PCOS, but the PCOS women with GDM were older and had higher BMI than the women without GDM. The mean BMI of our cohort of women with PCOS was 28.4 kg/m$^2$, markedly higher than previous studies on GDM in Norwegian cohorts [21,22], which might explain the high proportion of women with GDM in this study. Using the newer criteria for GDM, an even higher proportion of women was diagnosed with GDM, and it is noteworthy that the majority of these were diagnosed already in the first trimester. This increase in prevalence is mainly driven by a lower diagnostic limit of fasting glucose levels in the WHO 2013 criteria. Previous reports of unselected pregnant women indicate a stronger association between maternal BMI and fasting glucose than to glucose levels during an OGTT [23]. Also, in the present cohort of women with PCOS, both maternal age and BMI correlated stronger to fasting glucose level than to two-hour glucose level during an OGTT. In accordance with this, we observed that using the new WHO 2013 criteria revealed greater differences in maternal age and BMI between women with and without GDM.

## GDM effect on complications

In the present study, the clinical outcomes did not differ between PCOS women with and without GDM. This observation was independent of the GDM criteria used. However, women

with GDM in early pregnancy had a higher risk of late miscarriage; otherwise, having GDM did not associate with any maternal or neonatal complications such as development of pre-eclampsia or hypertension in pregnancy, or offspring birth weight.

This observation is noteworthy and contradicts the prevailing view that elevated glucose levels are the direct cause of pregnancy complications and poor pregnancy outcome. This finding is further supported by the observation that maternal BMI, but not glucose levels during pregnancy *per se*, correlated with birth weight z-score. Only a trend for univariate correlation with fasting glucose levels in the first trimester was observed. Importantly, and in contrast to maternal BMI, parity, age and gestational weight gain, having GDM was not a determinant for birth weight in the regression analysis. This importance of maternal BMI is in line with several previous reports demonstrating a positive association between maternal BMI and the risk of pregnancy complications such as preeclampsia and pregnancy outcomes such as increased infant birth weight and fat mass [23–27]. Several studies have also demonstrated that maternal pre-pregnancy BMI, independent of glucose levels, is an important risk factor for adverse pregnancy outcomes including macrosomia and LGA offspring [28–30]. The HAPO study reported a correlation between higher glucose levels and the prevalence of LGA offspring, but in the first publication of primary endpoint, birth size was not adjusted for maternal BMI [15]. More recent post hoc analyses of the HAPO study using the IADPSG criteria for GDM identified both maternal GDM and obesity at pregnancy week 28 (week 24–32) as independent risk factors for birth weight >90[th] percentile [31,32].

Our finding that gestational weight gain correlated independently to offspring birth weight is in accordance with previous studies on pregnant women with GDM [33,34]. Truong et al. demonstrated that a higher gestational weight gain associated with several adverse pregnancy outcomes including offspring LGA and preeclampsia, despite significantly lower prevalence of GDM among women with high weight gain [35]. In the first treatment studies of GDM which concluded that untreated women with GDM had increased risk for offspring LGA, the treatment group had lower gestational weight gain during pregnancy [13,14]. In these studies, the untreated women were not informed about their glucose levels and therefore received no lifestyle advice. Hence, it is difficult to discriminate the effect of the pharmacological treatment for GDM from the effects of diet and lifestyle modifications. In our cohort, the women diagnosed with GDM according to the WHO 1999 criteria had a markedly lower weight gain during pregnancy than did women without GDM. These women were diagnosed with GDM in the study period and received additional diet and lifestyle advice. However, when these women were excluded from the analysis, there was no difference in weight gain between women who were retrospectively classified with or without GDM with the new WHO 2013 criteria. Most women with GDM achieved acceptable glucose levels with only diet and lifestyle intervention and did not need additional treatment with insulin. Therefore, we hold that the reduced weight gain seen in women with GDM is a consequence of the GDM diagnosis and the diet and lifestyle treatment, and not by GDM *per se*.

Studies on dietary intervention in pregnant overweight women have shown a decrease in macrosomia with dietary intervention, despite similar gestational weight gain [36,37] and an association between the glycaemic load of the maternal diet and the prevalence of LGA offspring [38]. A recent RCT of overweight women without GDM in late second trimester found that intensive lifestyle intervention resulted in lower gestational weight gain during pregnancy, yet no reduction in the development of GDM; nevertheless, there was a markedly lower incidence of LGA offspring [39]. This is in concordance with our findings that in women with PCOS, gestational weight gain and not GDM status associated with offspring birth size, and that the risk of having an LGA offspring did not associate with GDM status. As increased maternal BMI and high gestational weight gain are risk factors for pregnancy complications,

we argue for an increased focus on overweight reduction and diet and lifestyle improvement in women with PCOS *prior to* conception. In addition, we suggest a focus on diet and gestational weight gain rather than strictly on GDM and glucose levels during pregnancy.

In the present study, having abnormal glucose metabolism already in the first trimester correlated to increased risk for late miscarriage. Since the insulin resistance due to pregnancy usually develops gradually from the second trimester, a pathological OGTT in the first trimester is compatible with insulin resistance and glucose intolerance prior to the actual pregnancy. However, in the regression analysis only 2-hour glucose level, and not fasting glucose, was a significant determinant for late miscarriage.

## Metformin effect

The previously published pooled analyses of this study cohort concluded that metformin reduced the incidence of late miscarriage and preterm birth (combined) in women with PCOS [16]. This conclusion is affirmed in the present post hoc study using more stringent diagnostic criteria for GDM. Furthermore, the effect is found to be independent of GDM status; however, the risk of late miscarriage was significantly lower only for metformin treated women with GDM according to WHO 2013 criteria, and in women without GDM according to WHO 1999 criteria. The mechanism is unclear, but probably not related to an effect on glucose metabolism since metformin treatment did not affect the proportion of women with GDM or the need for insulin treatment.

For almost all women in the present study, the WHO 1999 criteria were used for diagnosing GDM during the clinical studies. Although 200 women (28%) were diagnosed with GDM, and a large proportion of them in the first trimester, only 24 women (12% of the women with GDM) were treated with insulin, irrespective of metformin treatment. Despite an even higher GDM incidence when classifying the study cohort after the new WHO 2013 criteria, treatment with metformin did not reduce the GDM rate. This lack of metformin-effect on glucose homeostasis in pregnancy is remarkable as it has a well-known effect in non-pregnant women with PCOS, and on impaired glucose tolerance and diabetes type 2. However, this lack of effect of metformin on glucose metabolism in pregnant women has also been observed in previous RCTs on cohorts of overweight and obese pregnant women [40–43] and in a recent meta-analysis of RCTs on cohorts of women with increased GDM risk [44]. Importantly, the effect of metformin on glucose metabolism has not been tested in placebo controlled RCTs in women with GDM. We hold that women with PCOS, overweight and obese women with increased risk of GDM make up a significant proportion of women who develop GDM during pregnancy. Given this, metformin at best has a very limited effect on glucose metabolism in pregnant women. The use of metformin for the treatment of GDM should therefore be postponed until metformin has been shown to improve glucose homeostasis and/or pregnancy outcomes in placebo controlled RCTs of women with GDM.

## Strengths and limitations

The strengths of this study are the high number of participants included, where all women were diagnosed according to the same strict criteria before the actual pregnancy. This results in a well-defined population of pregnant women with PCOS. Further, the individual participant data analyses of three RCTs, the repeated longitudinal measurements of BMI and glucose levels, the meticulous collection of clinical information on co-morbidity, and strict registration and diagnoses of pregnancy outcomes add to the strengths of the study.

A main limitation of this study is that it is a post hoc analysis. In addition, the original studies included in the present analyses were not designed to primarily evaluate the effect of GDM

on pregnancy outcome. For instance, data on neonatal hypoglycaemia was not reliably recorded, and could not be included in the analyses. Another main limitation regarding the impact on GDM, is that this study included only women with PCOS. The results can therefore not be directly extrapolated to the general population of pregnant women. However, as PCOS is common among women of fertile age and since women with PCOS more often develop GDM, a relatively high proportion of the women with GDM can be expected to have PCOS. Our cohort of pregnant women with PCOS could therefore be representative for a significant proportion of pregnant women with GDM.

## Conclusion

In conclusion, in this large cohort of pregnant women with PCOS an additional GDM-diagnosis did not increase the risk for pregnancy complications except for an increased risk for late miscarriage among those with abnormal glucose metabolism already in the first trimester. Despite a lower maternal weight gain during pregnancy, metformin treatment neither affected the prevalence of GDM according to both old and new criteria for GDM, nor reduced the pregnancy complications in the women with GDM.

## Supporting information

**S1 Table. GDM in accordance with the Norwegian 2017-criteria in 685 women with PCOS.**
(DOCX)

**S2 Table. Pregnancy complications in patients treated with metformin and placebo, women with and without GDM (according to Norwegian 2017 criteria).**
(DOCX)

**S3 Table. Comparison of the prevalence between the different GMD classifications.** a. GDM diagnosed any time in pregnancy. b. GDM diagnosed in early pregnancy (at inclusion). (DOCX)

## Author Contributions

**Conceptualization:** Stine Lyngvi Fougner, Eszter Vanky, Tone Shetelig Løvvik, Sven Magnus Carlsen.

**Data curation:** Stine Lyngvi Fougner, Tone Shetelig Løvvik.

**Formal analysis:** Stine Lyngvi Fougner.

**Funding acquisition:** Eszter Vanky, Sven Magnus Carlsen.

**Investigation:** Stine Lyngvi Fougner, Eszter Vanky, Tone Shetelig Løvvik.

**Methodology:** Stine Lyngvi Fougner, Eszter Vanky, Tone Shetelig Løvvik, Sven Magnus Carlsen.

**Project administration:** Eszter Vanky, Tone Shetelig Løvvik, Sven Magnus Carlsen.

**Resources:** Eszter Vanky.

**Supervision:** Eszter Vanky, Sven Magnus Carlsen.

**Validation:** Stine Lyngvi Fougner, Tone Shetelig Løvvik, Sven Magnus Carlsen.

**Writing – original draft:** Stine Lyngvi Fougner.

**Writing – review & editing:** Eszter Vanky, Tone Shetelig Løvvik, Sven Magnus Carlsen.

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
