## [Decision Letter · Decision Letter 0]

25 Jan 2021

PONE-D-20-31750

The impact of gestational diabetes mellitus on pregnancy complications in women with PCOS

PLOS ONE

Dear Dr. Fougner,

Thank you for submitting your manuscript to PLOS ONE. After careful consideration, we feel that it has merit but does not fully meet PLOS ONE’s publication criteria as it currently stands. Therefore, we invite you to submit a revised version of the manuscript that addresses the points raised during the review process.

The reviewers have identified several importanct issues regarding both methodological and statistical concerns that need to be addressed

We look forward to receiving your revised manuscript.

Kind regards,

Stephen L Atkin, MD

Academic Editor

PLOS ONE

Journal Requirements:

2.Thank you for submitting your clinical trial to PLOS ONE and for providing the name of the registry and the registration number. The information in the registry entry suggests that your trial was registered after patient recruitment began. PLOS ONE strongly encourages authors to register all trials before recruiting the first participant in a study.

1) your reasons for your delay in registering this study (after enrolment of participants started);

2) confirmation that all related trials are registered by stating: “The authors confirm that all ongoing and related trials for this drug/intervention are registered.

3.We note that you have indicated that data from this study are available upon request. PLOS only allows data to be available upon request if there are legal or ethical restrictions on sharing data publicly. For information on unacceptable data access restrictions, please see http://journals.plos.org/plosone/s/data-availability#loc-unacceptable-data-access-restrictions.

4.Thank you for stating the following financial disclosure:

 "No

Reviewers' comments:

Reviewer's Responses to Questions

**Comments to the Author**

1. Is the manuscript technically sound, and do the data support the conclusions?

Reviewer #1: Yes

Reviewer #2: Partly

Reviewer #3: Partly

2. Has the statistical analysis been performed appropriately and rigorously? 

Reviewer #1: Yes

Reviewer #2: No

Reviewer #3: Yes

3. Have the authors made all data underlying the findings in their manuscript fully available?

Reviewer #1: Yes

Reviewer #2: No

Reviewer #3: Yes

4. Is the manuscript presented in an intelligible fashion and written in standard English?

Reviewer #1: Yes

Reviewer #2: Yes

Reviewer #3: Yes

5. Review Comments to the Author

Reviewer #1: This is an interesting paper, quite relevant to what are PLOS One's publication criteria.

It is becoming increasingly recognized that in GDM, maternal weight gain during the pregnancy is a major determinant of outcome, and may be more than important than hyperglycaemia.

The authors quite rightly make the point that no weight data was collected during the HAPO study, and is perfectly possible that the increased glucose levels associated with adverse outcomes may have been a very good correlate of increased body mass.

The authors have clearly demonstrated that in an (largely) obese and almost always insulin resistant group of pregnant women, given the label of PCOS, whatever level of new onset hyperglycaemia in pregnancy is found, then the outcome is no different. It is perhaps not surprising therefore then in those who were randomized to metformin there was little if any significant difference.

The major weakness of the study, acknowledged by the authors, is no control group of course.

Nonetheless I feel strongly that these findings add more to our understanding of the omportance of non-glycaemic factors in GDM.

Reviewer #2: Here is a list of specific comments. Note: line and page numbering in reviews and comments is based on ruler applied in Editorial Manager-generated PDF.

1. Page 7, lines 133–135: Per lines 106–107, data from the OGTT test at baseline were also available. I suggest including it in the Baseline Data section.

2. Page 8, lines 149–153: I did not understand what the z-scores represent (i.e., z-scores of what). I assumed it was birth weight. Did you mean to state the following? ‘For all live births after gestational week 24, a gestational age- and gender-adjusted birth weight was calculated based on Niklasson’s standard values from a large Swedish population. The z-score transformed birth weight express the deviation between observed birth weight and the Swedish population mean birth weight, adjusted for sex and gestational age at birth.’

3. Page 9, lines 160–162: In multivariable regressions, I suggest elaborating how determinants were selected into the regressions.

4. Page 10, line 183: Between the first and second paragraphs in the GDM and Baseline Characteristics section, it might be interesting to see a comparison between the WHO 1999 criteria and the WHO 2013 criteria using a 2x2 table with McNemar’s test statistics. Please feel free to add the Norwegian 2017 criteria to the mix but it certainly increases the number of comparisons.

5. Page 10, lines 195–198: I suggest specifying which correlation statistic was used (i.e., what R represented) in the Statistical Analyses section.

6. Page 11, line 221: In the logistic regressions in Table 2b, what was the rationale for not including GDM? Up to this point, all comparisons were centered around GDM. It would be interested to see how GDM associated with late miscarriage.

7. Page 13, line 257: The analyses in the Metformin Treatment in Women with and without GDM sections may be biased. The analyses were separated by GDM, a post Metformin variable. Analyses stratified by a post-baseline variable would require a more careful analytic approach or require a cautious note regarding the potential bias the analyses may infer.

8. Table 2b:

(8a) For both univariable and multivariable logistic regressions, I suggest reporting odds ratios and their 95% confidence intervals (instead of B). Also, I suggest replacing “multivariate” with ‘multivariable’.

(8b) Lines 569–570: For continuous variables, t-test was not appropriate in this case. I suggest reporting p-values using univariable logistic regressions. Note that the univariable logistic regressions can be used for dichotomous variables as well where p-values should be the same as the p-values from chi-square tests.

9. Table 3:

(9a) Please clarify what R represented.

(9b) Instead of beta/t, I suggest reporting regression coefficients (betas) and their 95% confidence intervals.

Reviewer #3: I have reviewed this submission by Fougner et al. The study included three cohorts of PCOS women randomized in three different trials in three different eras to receive metformin versus placebo. The study is comparing the pregnancy outcomes between women with and without GDM. The study has significant issues that question it's validity.

Major points

1. The study title indicates that the focus of the paper is on pregnancy outcomes. However, the manuscript includes lengthy details describing the difference in women's prevalence and characteristics with GDM using three different criteria. In my view, this could be the main focus of the manuscript. The title should reflect this part of the study.

2. The authors did not outline what the three diagnostic criteria in the manuscript are. In particular – what is the Norwegian criteria?

3. There was no standardized management protocol for women with GDM. Hence, it cannot be assumed that all women were treated to a unified target. The glycaemic management of women with GDM is quite critical to pregnancy outcomes.

4. The authors used different criteria and applied them to the whole cohort and then classified them as GDM or no GDM based on three different criteria. However, this is a messy and untidy way to define GDM. It essentially means that some women classified with GDM using one classification were essentially classified as Normal Glucose Tolerant ( NGT) during pregnancy and did not receive any treatment.

5. Glycaemic control is the most critical factor in the pregnancy outcomes of GDM. There was neither unified management protocol nor consistency in the women's classification during pregnancy- hence the pregnancy outcomes are not valid.

6. Hence, If the authors would like to proceed with this paper, I will advise them to drop the pregnancy outcomes and only report the prevalence of GDM using three different criteria. Alternatively, they can do a re-analysis using the actual classification of the women during pregnancy.

Minor points

1. Under the abstract, the authors wrote, "having GDM according to newer criteria correlated to increased maternal age and BMI (p<0.001), while GDM already in the first trimester associated with increased risk for late miscarriage (p<0.01)." What does this mean?

2. In the introduction, the authors wrote "IFD diabetes atlas" is IDF- diabetes atlas.

3. They stated, "The HAPO study demonstrated a positive association between glucose levels and the proportion of neonates with a birth weight above 4000 g". This is not correct; the HAPO showed correlation with birth weight > 90th percentile.

4. The HAPO trial reported on Gestational weight gain. The authors can refer to some of their publications.

5. The authors should report all the p-values and not only mention ns.

6. PLOS authors have the option to publish the peer review history of their article (what does this mean?). If published, this will include your full peer review and any attached files.

Reviewer #1: **Yes: **Stephen Beer

Reviewer #2: No

Reviewer #3: **Yes: **Mohammed Bashir

---

## [Author Response · Author response to Decision Letter 0]

5 Apr 2021

We thank the Editor and the Reviewers for their time and effort spent on reviewing the manuscript. Below are the responses to the comments by each of the reviewers.

Reviewer #1: This is an interesting paper, quite relevant to what are PLOS One's publication criteria. It is becoming increasingly recognized that in GDM, maternal weight gain during the pregnancy is a major determinant of outcome, and may be more than important than hyperglycaemia.

The authors quite rightly make the point that no weight data was collected during the HAPO study, and is perfectly possible that the increased glucose levels associated with adverse outcomes may have been a very good correlate of increased body mass.

The authors have clearly demonstrated that in an (largely) obese and almost always insulin resistant group of pregnant women, given the label of PCOS, whatever level of new onset hyperglycaemia in pregnancy is found, then the outcome is no different. It is perhaps not surprising therefore then in those who were randomized to metformin there was little if any significant difference.

The major weakness of the study, acknowledged by the authors, is no control group of course.

Nonetheless I feel strongly that these findings add more to our understanding of the importance of non-glycaemic factors in GDM.

Answer: Thank you for thorough reading of our manuscript and your positive review.

We appreciate your comment on the importance of gaining more knowledge on the factors not related to hyperglycemia in women with GDM, which we now believe to be more important than previously regarded.

Reviewer #2: Here is a list of specific comments. Note: line and page numbering in reviews and comments is based on ruler applied in Editorial Manager-generated PDF.

1. Page 7, lines 133–135: Per lines 106–107, data from the OGTT test at baseline were also available. I suggest including it in the Baseline Data section.

Answer: Thank you for your suggestion. We agree and have included the information in the Baseline Data section.

2. Page 8, lines 149–153: I did not understand what the z-scores represent (i.e., z-scores of what). I assumed it was birth weight. Did you mean to state the following? ‘For all live births after gestational week 24, a gestational age- and gender-adjusted birth weight was calculated based on Niklasson’s standard values from a large Swedish population. The z-score transformed birth weight express the deviation between observed birth weight and the Swedish population mean birth weight, adjusted for sex and gestational age at birth.’

Answer: Thank you for pointing out some unclarity in our text. We have now adjusted the text as suggested.

3. Page 9, lines 160–162: In multivariable regressions, I suggest elaborating how determinants were selected into the regressions.

Answer: Thank you for your correct notification. This is now included in the Statistics section.

4. Page 10, line 183: Between the first and second paragraphs in the GDM and Baseline Characteristics section, it might be interesting to see a comparison between the WHO 1999 criteria and the WHO 2013 criteria using a 2x2 table with McNemar’s test statistics. Please feel free to add the Norwegian 2017 criteria to the mix but it certainly increases the number of comparisons.

Answer: We agree and have added this 2x2 table in the Supplemental file (Suppl. Table 3). 

5. Page 10, lines 195–198: I suggest specifying which correlation statistic was used (i.e., what R represented) in the Statistical Analyses section.

Answer: Thank you for the notification of omitted important information. This is now included in the Statistics section

6. Page 11, line 221: In the logistic regressions in Table 2b, what was the rationale for not including GDM? Up to this point, all comparisons were centered around GDM. It would be interested to see how GDM associated with late miscarriage.

Answer: For this regression analysis, we wanted to include the glucose data (fasting glucose and 2h-glucose) instead of the dichotomized parameter GDM. This was possible since the glucose data were collected at one time point in pregnancy. We found it more interesting to analyze if fasting glucose was more important than 2h-glucose, or vice versa, than using the “double” dichotomized parameter GDM. Using continuous data (fasting and 2 h glucose) instead of dichotomized data (GDM) in general also increase the statistical power and possibility to identify possible biologic associations. For the regression analysis of birth weight, however, we had glucose data from different time points in the pregnancy, and this was not possible. For this analysis, we had to use the dichotomized parameter GDM sometime in the pregnancy and perform two analyses, one for each GDM criterion (WHO 1999 and WHO 2013).

7. Page 13, line 257: The analyses in the Metformin Treatment in Women with and without GDM sections may be biased. The analyses were separated by GDM, a post Metformin variable. Analyses stratified by a post-baseline variable would require a more careful analytic approach or require a cautious note regarding the potential bias the analyses may infer.

Answer: For the analyses of parameters registered late on pregnancy or at birth like preeclampsia and, we agree with the reviewer. There is a potential bias when looking at the GDM and non-GMD group separately, where we cannot be sure if the treatment with Metformin during second and third trimester influence the later development of GDM. However, we did not find any effect of Metformin regarding the glucose levels or the prevalence of GDM. There is still a theoretically risk for Metformin influencing which women who develop GDM, however, we hold this as less possible. In this regard, it is interesting that the observed effects of metformin are similar for both groups (GDM and non-GMD). However, for birth weight, this was accounted for in the regression analyses where both Metformin and GDM status were included (please see page 12).

In addition, regarding the risk for late miscarriage and preterm birth combined, the risk for this potential bias is even less likely, since most women with GDM in this group were diagnosed at inclusion. Of the 6 women diagnosed with GDM later in pregnancy after any criteria, three had received Metformin and three placebo treatment. We have not included these details in the paper, but we can do so if indicted by Editor.

8. Table 2b:

(8a) For both univariable and multivariable logistic regressions, I suggest reporting odds ratios and their 95% confidence intervals (instead of B). Also, I suggest replacing “multivariate” with ‘multivariable’.

Answer: Thank you for the correct notification on parameters reported. The Table is now updated.

(8b) Lines 569–570: For continuous variables, t-test was not appropriate in this case. I suggest reporting p-values using univariable logistic regressions. Note that the univariable logistic regressions can be used for dichotomous variables as well where p-values should be the same as the p-values from chi-square tests.

Answer: Thank you for the correct notification on parameters reported. We have now reported the p-values using univariable logistic regression, in addition to reporting odds ratio and confidence interval (except for one parameter where confidence interval did not give any sense – please see revised Table legend).

9. Table 3:

(9a) Please clarify what R represented.

Answer: Thank you for the correct notification on parameters reported. R represented the Pearson correlation coefficient but is now replaced with beta and confidence interval from univariable linear regression. 

(9b) Instead of beta/t, I suggest reporting regression coefficients (betas) and their 95% confidence intervals. 

Answer: Thank you for the correct notification on parameters reported. The Table is now updated.

Reviewer #3: I have reviewed this submission by Fougner et al. The study included three cohorts of PCOS women randomized in three different trials in three different eras to receive metformin versus placebo. The study is comparing the pregnancy outcomes between women with and without GDM. The study has significant issues that question it's validity.

Major points

1. The study title indicates that the focus of the paper is on pregnancy outcomes. However, the manuscript includes lengthy details describing the difference in women's prevalence and characteristics with GDM using three different criteria. In my view, this could be the main focus of the manuscript. The title should reflect this part of the study.

Answer: We hold the view that the title should reflect our hypothesis (the reason for performing the study) and/or the most important observation. Our main focus of this study was to evaluate the impact of GDM on pregnancy complications. However, to do this in detail, we had to classify the patients according to the criteria for GDM which has recently been changed. Our opinion is that an analysis with only old and now outdated criteria would not be sufficient. With our almost complete glucose data we choose to evaluate this question using the newer GDM criteria now widely used, in addition to the older criteria used at the time of the studies. At the same time, we had the opportunity to compare both the old and the new GDM criteria to pregnancy outcome in this large, well defined patient cohort and also to evaluate the possible difference between the GDM criteria. We state that pregnancy outcome is the most important part of our publication, and that the title therefore is correct. We consider the prevalence of GDM and the clinical characteristics of women with GDM according to the different criteria more to be a necessary and interesting introduction to analysis of the main hypothesis in this work. We are not aware that any data similar to our data on the missing association between glucose levels and pregnancy outcomes has been presented before. That is the novelty of the present study and not the difference in GDM according to different criteria. However, we have adjusted the title, now indicating the use of different CGM criteria.

2. The authors did not outline what the three diagnostic criteria in the manuscript are. In particular – what is the Norwegian criteria?

Answer: Information on all GDM criteria used were, and still are, stated in the Methods section, under the subsection GDM Classification. No adjustments of the manuscript have been made at this point.

3. There was no standardized management protocol for women with GDM. Hence, it cannot be assumed that all women were treated to a unified target. The glycaemic management of women with GDM is quite critical to pregnancy outcomes.

Answer: All women were classified as having GDM and treated for GDM according to national standard and guidelines at the time. Thus, none were deprived of standard GDM treatment. 

We agree that the management of GDM are important, but our data on missing association between glucose levels and pregnancy outcomes in a well-defined cohort of pregnant women with PCOS challenge the view that glycaemic management is most critical. 

4. The authors used different criteria and applied them to the whole cohort and then classified them as GDM or no GDM based on three different criteria. However, this is a messy and untidy way to define GDM. It essentially means that some women classified with GDM using one classification were essentially classified as Normal Glucose Tolerant (NGT) during pregnancy and did not receive any treatment.

Answer: All women were classified as having GDM and treated for GDM according to national guidelines, thus, no women were deprived of standard GDM treatment. For almost all included women, the same GDM classification and clinical practice for management were used (WHO 1999 criteria). Only very few of the women in the last study (PregMet 2), that is only the last 10-15 included women in Iceland, were classified according to the new WHO 2013 criteria. Of them, only 3 women were diagnosed with GDM after new criteria, and only one patient received insulin treatment (from pregnancy week 36).

We hold that the fact that some women in these studies, according to criteria developed after their pregnancy, would have been classified as having GDM today, or vice versa, do not change the importance of these evaluations. Since almost all women in these studies were included prior to implementation of the newer GDM criteria, we also had the possibility to evaluate the group of women that did not get a GDM diagnose in pregnancy (according to the old GDM criteria) and therefore no treatment, but would have been diagnosed with GDM today, according to the new criteria. In this way, we argue that differences in gestational age and gestational weight gain is a result of the treatment and not the condition itself. 

5. Glycaemic control is the most critical factor in the pregnancy outcomes of GDM. There was neither unified management protocol nor consistency in the women's classification during pregnancy- hence the pregnancy outcomes are not valid.

Answer: We agree that the common wisdom hold by most physicians is that glycemic control is the most critical factor influencing on outcomes in GDM. However, this is what our data and our paper challenge. Regarding diagnose and management of GDM, see answers above. 

6. Hence, If the authors would like to proceed with this paper, I will advise them to drop the pregnancy outcomes and only report the prevalence of GDM using three different criteria. Alternatively, they can do a re-analysis using the actual classification of the women during pregnancy.

Answer: With all respect, we do not agree with this suggestion. We find it important to study the impact of glucose levels and the presence of gestational diabetes on pregnancy outcomes in a well-defined patient cohort. The fact that our data challenge the previous opinion that glucose levels are the most critical factor influencing outcome, makes it important to publish our data and make the findings available. 

In fact, the analyses of GDM according to the WHO 1999 criteria is the analyses using the actual classification of the women used during pregnancy, as the Reviewer suggest, please see the answer to pt. 4 above. Together with other studies, our data might lead to an increased focus on the total risk profile of the pregnant women and not only on the glucose levels. 

Minor points

1. Under the abstract, the authors wrote, "having GDM according to newer criteria correlated to increased maternal age and BMI (p<0.001), while GDM already in the first trimester associated with increased risk for late miscarriage (p<0.01)." What does this mean?

Answer: Having GDM anytime during pregnancy correlated to increased maternal age and BMI. Having GDM already at inclusion in the first trimester associated with risk of late miscarriage. For the women with late miscarriage, only glucose data from the inclusion in the late first trimester was available. We have rephrased the sentences and hope this is clearer now.

2. In the introduction, the authors wrote "IFD diabetes atlas" is IDF- diabetes atlas.

Answer: Thank you for pointing out our misspelling. This has been corrected.

3. They stated, "The HAPO study demonstrated a positive association between glucose levels and the proportion of neonates with a birth weight above 4000 g". This is not correct; the HAPO showed correlation with birth weight > 90th percentile.

Answer: The original publication from the HAPO study, ref. 15, found an association with birth weight above 4000 g, but a newer post hoc analysis of the HAPO material correctly found a correlation with birth weight >90th percentile. This is already discussed in the Discussion section, and with ref. 31.

4. The HAPO trial reported on Gestational weight gain. The authors can refer to some of their publications.

Answer: We have both read and referred to several of the HAPO publications. However, we have not been able to find any publication referring data on maternal gestational weight gain. We will be happy to discuss and refer a publication from this large and important study reporting weight data if there are publications that we have not found. Please, see also the comment made by Reviewer #1 on this topic.

5. The authors should report all the p-values and not only mention ns.

Answer: We agree and have added the exact p-value to the Table.

---

## [Decision Letter · Decision Letter 1]

30 Apr 2021

PONE-D-20-31750R1

No impact of gestational diabetes mellitus on pregnancy complications in women with PCOS, regardless of GDM criteria used.

PLOS ONE

Dear Dr. Fougner,

Thank you for submitting your manuscript to PLOS ONE. After careful consideration, we feel that it has merit but does not fully meet PLOS ONE’s publication criteria as it currently stands. Therefore, we invite you to submit a revised version of the manuscript that addresses the points raised during the review process.

Please address the comments of the reviewers particularly on the queries raised on the methodology by reviewer 2

We look forward to receiving your revised manuscript.

Kind regards,

Stephen L Atkin, MD

Academic Editor

PLOS ONE

Reviewers' comments:

Reviewer's Responses to Questions

**Comments to the Author**

1. If the authors have adequately addressed your comments raised in a previous round of review and you feel that this manuscript is now acceptable for publication, you may indicate that here to bypass the “Comments to the Author” section, enter your conflict of interest statement in the “Confidential to Editor” section, and submit your "Accept" recommendation.

Reviewer #1: (No Response)

Reviewer #4: (No Response)

2. Is the manuscript technically sound, and do the data support the conclusions?

Reviewer #1: Yes

Reviewer #4: Yes

3. Has the statistical analysis been performed appropriately and rigorously? 

Reviewer #1: Yes

Reviewer #4: Yes

4. Have the authors made all data underlying the findings in their manuscript fully available?

Reviewer #1: Yes

Reviewer #4: Yes

5. Is the manuscript presented in an intelligible fashion and written in standard English?

Reviewer #1: No

Reviewer #4: Yes

6. Review Comments to the Author

Reviewer #1: Having reviewed the submission carefully, and the comments of reviewer 3 in particular, I do still feel that this study should be published. I think it would be a better paper if it was presented as a study of different classifications of GDM, but the authors are being perfectly reasonable in sticking to their original hypothesis. I think they have justified this decision, and as such satisfy, in their revisions, much of the original concerns. I do have one or two major issues though.

I apologise to the Editor and the authors. The HAPO study does present data on weight in pregnancy in GDM, but not weight gain. Hyperglycaemia and Adverse pregnancy outcome (HAPO) study: Association with maternal body mass index BJOG 2010 117(5): 577-84

This concludes 'higher maternal BMI, independently of maternal glycaemia is strongly associated with increased frequency of pregnancy complications'.

Secondly, The Hyperglycaemia and Adverse Pregnancy outcome Study. Association of GDM and obesity with pregnancy outcomes. Diabetes Care 2012 35(4): 780-786 which concludes 'both maternal GDM and obesity are independently associated with adverse pregnancy outcomes.

These references should be included.

I think is incorrect to state that metformin use should be suspended until placebo controlled RCTs are performed in GDM, this isn't going to happen.

Finally as a minor point there is a tangle in the use of English, line 164 it should say parameters that either were instead of either was and in line 166 it should say only one was rather than only one were.

Reviewer #4: Title: No impact of gestational diabetes mellitus on pregnancy complications in women with PCOS, regardless of GDM criteria used.

Authors: Fougner et al.

Manuscript ID: PONE-D-20-31750R1

I accepted to review the paper by Fougner et al. in second revision.

The authors in this subanalysis of three RCTs concluded that, in pregnant women with PCOS, the GDM diagnosis did not increase the risk of late pregnancy complications. This data is surprising and it may be explained in only two ways: the risk of pregnancy complications in women with PCOS is per se so high that it is difficult to increase it and/or pregnant women with PCOS have different response to GDM in comparison with non-PCOS populations.

The introduction section should be reduced in length of about 30%.

The authors define the GDM “criteria” using only glucose values at OGTT irrespective from gestational age (when assays were taken). Please complete the methods section and discuss.

Have you excluded patients with an abnormal OGTT at 12 weeks? Please clarify and discuss.

Can you give data on pregnancy complications on both populations (with GDM and pregestational diabetes)?

Please clarify if all patients had a natural pregnancy or ART pregnancy (including medications/drugs).

Please clarify where the patients were followed. One of the main concerns in terms of pregnancy complications in infertile e/o PCOS patients is the Hospital for obstetric management.

In my opinion, the analysis should be restricted to PCOS patients who did not use metformin (non-randomized to metformin) in order to avoid a potential confounder. Data on metformin should be analyzed as secondary aim in another different population and substudy.

The authors should clarify how the maternal weight gain was calculated and analyzed as risk factor. The increase in body weight throughout pregnancy is physiologic and different in obese and non-obese women (suggestions of the international guidelines).

7. PLOS authors have the option to publish the peer review history of their article (what does this mean?). If published, this will include your full peer review and any attached files.

Reviewer #1: **Yes: **Stephen Beer

Reviewer #4: No

---

## [Author Response · Author response to Decision Letter 1]

14 Jun 2021

Dear Editor,

Thank you for considering our manuscript “No impact of gestational diabetes mellitus on pregnancy complications in women with PCOS, regardless of GDM criteria used” for publication in PLOS ONE.

We thank the Editor and the Reviewers for their time and effort spent on reviewing the manuscript. Below are the responses to the comments by each of the reviewers.

Reviewer #1: Having reviewed the submission carefully, and the comments of reviewer 3 in particular, I do still feel that this study should be published. I think it would be a better paper if it was presented as a study of different classifications of GDM, but the authors are being perfectly reasonable in sticking to their original hypothesis. I think they have justified this decision, and as such satisfy, in their revisions, much of the original concerns. I do have one or two major issues though.

I apologise to the Editor and the authors. The HAPO study does present data on weight in pregnancy in GDM, but not weight gain. Hyperglycaemia and Adverse pregnancy outcome (HAPO) study: Association with maternal body mass index BJOG 2010 117(5): 577-84. 

This concludes 'higher maternal BMI, independently of maternal glycaemia is strongly associated with increased frequency of pregnancy complications'.

Secondly, The Hyperglycaemia and Adverse Pregnancy outcome Study. Association of GDM and obesity with pregnancy outcomes. Diabetes Care 2012 35(4): 780-786 which concludes 'both maternal GDM and obesity are independently associated with adverse pregnancy outcomes.

These references should be included.

Answer: We agree that data on maternal weight measured in third trimester have been published in later publications from the HAPO study. The study published in Diabetes Care 2012 was already included and discussed in the Discussion section (reference 31). We have now included also the HAPO publication in BJOG 2010 (new reference 32).

I think is incorrect to state that metformin use should be suspended until placebo controlled RCTs are performed in GDM, this isn't going to happen.

Answer: We agree that this maybe is not going to happen, at least worldwide. However, our opinion is still that further studies, RCTs, on the efficacy of metformin is necessary before treating pregnant women with a medication passing the placenta and being present in fetal blood at the same level as in the mother. Several studies suggest that metformin-exposed children have higher BMI and more often overweight during childhood (Hanem JCEM 2018, Rowan BMJ Open Diab Res Care 2018). RCTs on the efficacy of metformin treatment in pregnant women are now only available in women with PCOS, women with hypertension, women with insulin resistance prior to pregnancy and obese women. In these studies, despite the high risk for GDM in these women, metformin treatment did not influence the glucose homeostasis nor reduced the incidence of GDM.

Finally as a minor point there is a tangle in the use of English, line 164 it should say parameters that either were instead of either was and in line 166 it should say only one was rather than only one were.

Answer: We are sorry for the grammatical errors. Thank you for noticing, they are now corrected.

Reviewer #4: The authors in this subanalysis of three RCTs concluded that, in pregnant women with PCOS, the GDM diagnosis did not increase the risk of late pregnancy complications. This data is surprising and it may be explained in only two ways: the risk of pregnancy complications in women with PCOS is per se so high that it is difficult to increase it and/or pregnant women with PCOS have different response to GDM in comparison with non-PCOS populations.

Answer: We agree that our data could be surprising, however, as discussed in the Introduction and the Discussion sections other studies have also suggested that underlying factors can be more important than the glucose level per se. We agree that the two mentioned explanations both are possible. We cannot know whether women with PCOS have a “different GDM” or a different response to GDM compared to women without PCOS, since all women in these studies have PCOS. But as PCOS is a common condition and the prevalence of GDM is markedly increased in PCOS, these women probably constitute a significant proportion of all women with GDM. This is discussed at the end of the Discussion section.

The introduction section should be reduced in length of about 30%.

Answer: We feel that all parts of the introduction now is necessary to explain our rationale for this substudy. The length of the introduction has not been commented by the other three reviewers. However, we will try to shorten the section if the Editor find it best.

The authors define the GDM “criteria” using only glucose values at OGTT irrespective from gestational age (when assays were taken). Please complete the methods section and discuss.

Answer: As described in the Methods section, an OGTT was performed after protocol at two or three different visits during the pregnancy. We classified GDM using the glucose levels at each time point. It is correct that we use the same glucose levels at the different study visits.

Have you excluded patients with an abnormal OGTT at 12 weeks? Please clarify and discuss.

Can you give data on pregnancy complications on both populations (with GDM and pregestational diabetes)?

Answer: Patients with known pregestational diabetes and with fasting glucose ≥7 mmol/l at screening were not included in the randomized studies (exclusion criteria – please see the referred original publications). Women included in the RCTs that had an abnormal OGTT at the inclusion visit at appr. 12 weeks were not excluded in this substudy. They were registered as having early GDM. For data on pregnancy complications in the population with abnormal versus normal OGTT at 12 weeks, please see the sections “GDM at inclusion” and “Early versus late GDM”.

Please clarify if all patients had a natural pregnancy or ART pregnancy (including medications/drugs).

Answer: Of the 722 women included in this substudy, 312 women (43 %) had a spontaneous pregnancy, while the remaining 410 women had received assistance. Detailed information on this is given in the original publication for each RCT.

Please clarify where the patients were followed. One of the main concerns in terms of pregnancy complications in infertile e/o PCOS patients is the Hospital for obstetric management.

Answer: The women were followed with regular clinical and study visits at the Department of Gynaecology and Obstetrics of each study hospital. Detailed information on this is given in the original publication for each RCT.

In my opinion, the analysis should be restricted to PCOS patients who did not use metformin (non-randomized to metformin) in order to avoid a potential confounder. Data on metformin should be analyzed as secondary aim in another different population and substudy.

Answer: Metformin could be a potential confounder. However, we did not find any difference between the women treated with metformin or not, except for the combined endpoint late miscarriage and preterm birth. Particularly, there were absolutely no difference in glucose levels or GDM incidence between the groups receiving metformin or placebo. This question is also taken into account with the analyses done separately for the two groups (metformin and placebo), see table 5, and also in the regression analyses where randomization was included as variable. Also, for the analyses of early GDM, the women had not yet started with study medication. 

The authors should clarify how the maternal weight gain was calculated and analyzed as risk factor. The increase in body weight throughout pregnancy is physiologic and different in obese and non-obese women (suggestions of the international guidelines).

Answer: As described in the Methods section, Pregnancy outcome, maternal weight gain was calculated as the difference between maternal weight at the visit in pregnancy week 36 and the maternal weight at the inclusion visit in week 12, given in kilogram. We did not relate the weight gain in kg to the IOM guideline according to prepregnant maternal BMI.

---

## [Decision Letter · Decision Letter 2]

7 Jul 2021

No impact of gestational diabetes mellitus on pregnancy complications in women with PCOS, regardless of GDM criteria used.

PONE-D-20-31750R2

Dear Dr. Fougner,

We’re pleased to inform you that your manuscript has been judged scientifically suitable for publication and will be formally accepted for publication once it meets all outstanding technical requirements.

Kind regards,

Stephen L Atkin, MD

Academic Editor

PLOS ONE

Additional Editor Comments (optional):

Reviewers' comments:

Reviewer's Responses to Questions

**Comments to the Author**

1. If the authors have adequately addressed your comments raised in a previous round of review and you feel that this manuscript is now acceptable for publication, you may indicate that here to bypass the “Comments to the Author” section, enter your conflict of interest statement in the “Confidential to Editor” section, and submit your "Accept" recommendation.

Reviewer #1: All comments have been addressed

Reviewer #4: All comments have been addressed

2. Is the manuscript technically sound, and do the data support the conclusions?

Reviewer #1: Yes

Reviewer #4: Yes

3. Has the statistical analysis been performed appropriately and rigorously? 

Reviewer #1: Yes

Reviewer #4: Yes

4. Have the authors made all data underlying the findings in their manuscript fully available?

Reviewer #1: Yes

Reviewer #4: Yes

5. Is the manuscript presented in an intelligible fashion and written in standard English?

Reviewer #1: Yes

Reviewer #4: Yes

6. Review Comments to the Author

Reviewer #1: (No Response)

Reviewer #4: The manuscript has been improved. The authors have followed all suggestions and replied to all queries.

7. PLOS authors have the option to publish the peer review history of their article (what does this mean?). If published, this will include your full peer review and any attached files.

Reviewer #1: **Yes: **Stephen Beer

Reviewer #4: **Yes: **Prof. Stefano Palomba

---

## [Editor Report · Acceptance letter]

14 Jul 2021

PONE-D-20-31750R2 

No impact of gestational diabetes mellitus on pregnancy complications in women with PCOS, regardless of GDM criteria used. 

Dear Dr. Fougner:

I'm pleased to inform you that your manuscript has been deemed suitable for publication in PLOS ONE. Congratulations! Your manuscript is now with our production department. 

Kind regards, 

on behalf of

Dr. Stephen L Atkin 

Academic Editor

PLOS ONE